# Environmental Pollution Liability Insurance Pricing and the Solvency of Insurance Companies in China: Based on the Black–Scholes Model

**DOI:** 10.3390/ijerph20021630

**Published:** 2023-01-16

**Authors:** Shuai Chen, Jiameng Yang

**Affiliations:** 1Business School, University of Shanghai for Science and Technology, Shanghai 200093, China; 2College of Economics and Management, Nanjing Forestry University, Nanjing 210037, China

**Keywords:** environmental pollution liability insurance, pricing, solvency, insurance company, Black–Scholes model

## Abstract

Environmental pollution liability insurance is becoming increasingly important for China to achieve its emission reduction targets. Insurance pricing is a crucial factor restricting the market share of environment pollution liability insurance, from the perspective of the Black–Scholes pricing model, which in turn has influenced the solvency of insurance companies in China. Firstly, this study analyzes the problems existing in compulsory liability insurance for environmental pollution in China. It proceeds with analyzing the price of compulsory environmental pollution liability insurance using the Black–Scholes pricing model, and derives a high premium insurance rate of 2.44%. Moreover, it performs a multivariate regression analysis using the asset and liability data, taken from the annual report, to identify three key factors affecting the solvency adequacy ratio, namely, capital debt ratio, reflecting the company asset structure; net interest rate on assets, reflecting the asset scale with actual solvency; and claim ratio, reflecting the business quality. Based on the results of regression analysis and robustness test for the China Insurance Clauses (CIC) company, People’s Insurance Company of China (PICC), and Asia-Pacific Property & Casualty Insurance (API) company, it is shown that the effect of total asset, total debt, capital debt ratio, claim ratio, and net interest rate on assets on the solvency adequacy ratio is significant, with respect to the size of the coefficients. Based on the Black–Scholes pricing model found in the previous cycle of liability insurance, and keeping in view the existing problems of environmental pollution liability insurance expenditure, this paper presents suggestions that are conducive to improving the solvency of insurance companies in China.

## 1. Introduction

Increasing environmental protection expenditure is important for carbon dynamics and to maintain the ecological and environmental balance in China [1]. Both green insurance subsidies and government subsidies promote firms’ innovation, but a green insurance subsidy owns lower risk than a direct subsidy to innovate [2]. In these industries, entities might have stronger incentives to invest in leading time reduction and in information acquisition.

### 1.1. Research Motivation

Energy Performance Contracting (EPC) is effective in expanding channels of information and reducing transaction and implementation costs. EPC is a market-based energy-saving mechanism. From 2006 to 2010, investments in EPC projects increased approximately 10 times [3]. China had 782 Energy Service Companies (ESCOs) in 2010 [4]; in addition, EPC involves a large amount of investment, long cycle, and high return on investment. Due to the limited investment capacity of economic, social, and cultural organizations, bank loans have become the main source of funding EPC [5]. According to the “shared savings contract”, an ESCO is responsible for shared savings [3]. The shared savings contract is suitable for countries that transfer risks, such as developing countries. Shared savings contracts are widely used in China; this is partially due to the decline in the EU’s willingness to invest in energy dialogues. Moreover, the government’s support is also an essential reason why shared contracts are widely used in China. The empirical study in [6] found that energy-saving levels are highly uncertain. After investigating the market performance of 63 ESCOs in the US, we found that in the 369 survey responses, 57% of the projects achieved better energy savings than expected, 30% were lower than expected, and only 13% were as expected. The uncertainty of the energy-saving levels of these projects directly leads to the uncertainty of single-stage energy savings. Due to the uncertainty of energy savings, the current energy service industry in China is in a marginal period. The energy-saving measures and indicators reflect the importance of energy-saving and emission reduction insurance products.

The failure of personnel and technology to keep up with the development of the EPC market is a big obstacle [7]. According to a case study of the Norwegian municipal authority, determined and decisive decision-makers in the public sector can promote EPC to find better solutions and increase the possibility of outsourcing. The three-model energy-saving service companies that were established after the implementation of the World Bank–GEF Energy Saving Promotion Project in 1997 include Yuanshen Energy Saving Technology Co., Ltd. of Beijing, China, Energy Saving Technology Co., Ltd. of Shenyang, China, and Energy Saving Engineering Co., Ltd. of Jinan, China [8]. From 1997 to 2006, the efficiency of energy consumption increased steadily. Up to 475 energy-saving projects have been implemented in 405 enterprises, with a total investment of more than CNY 133.1 million. Through these projects, ESCOs obtained a net profit of CNY 420 million; the net income of customers is equivalent to 8 to 10 times that of the ESCOs. The energy-saving levels of these projects reached 15.1 billion tons of standard coal, and the resulting carbon emissions reduced to 1.45 million tons per year, indicating that EPC has sufficient capacity to develop in China.

Climate change is a risk that insurance companies cannot cover [9], but environmental risks meet the insurer’s underwriting conditions. Adam Whitmore [10] analyzed the climate change problem caused by environmental pollution, and he applied the externality model to the climate change problem and proposed a solution that combines compulsory liability insurance and government measures to mitigate these problems. Lockett Nick made a comparative analysis of insurance coverage in Europe and the US and proposed corresponding countermeasures [11]. The dynamic changes in industrial enterprises’ demand for environmental pollution liability insurance is analyzed by studying the emission reduction system of enterprise risk management and other types of enterprise management in environmental protection [12]. Regarding model selection research, insurance models for environmental liability insurance in various parts of the world and China include mandatory and voluntary insurance models. The effective resolution of environmental pollution victims’ claims requires that environmental liability insurance is legislatively enforced [13]. The US and Sweden are typical of compulsory environmental liability insurance. The insurance coverage and protection of compulsory environmental liability insurance is considered in the US in [14]. The insurance coverage and insurance models of environmental pollution liability in the US and the UK is compared in [15]. In the UK and France, voluntary insurance is the main model. Industries under special laws and regulations, such as oil, mining, and nuclear energy, have adopted a mandatory model. Feng Yan et al. conducted a comparative case study between voluntary (Chongqing City) and mandatory (Wuxi City, Jiangsu Province) pollution insurance to assess local pollution insurance practices in China [16]. They explained the differences in policies, pollution control, and attitudes of companies with reduced emissions in the development of the pollution insurance market in Chongqing and Wuxi. They believed that compulsory environmental liability insurance is essential for the government to establish a comprehensive energy-saving emission reduction insurance system.

Regarding system construction research, Ji Maggie (2012) posited that there are two types of insurance policies, which are independent environmental pollution liability and public liability insurance [17]. A detailed introduction to the application of the green insurance policy model in the US, its development background, and its impact on emission reduction are given [18]. Moreover, the risks and scope of underwriting are discussed. John Merrifield believed that risk transfer is the fundamental reason for the existence and development of environmental liability insurance. The risk of environmental accident compensation requires that the insured shares the damage compensated for [19]. The practice of environmental liability insurance is analyzed, and for green insurance to be effective, it should be compulsory liability insurance [20]. The current development status of the green insurance business in some countries, such as the US and the UK, as well as the EU, is discussed in [21]. The success of green insurance is attributed to the joint underwriting of insurance companies that have designed many new policies to suit the market demand. A variety of environmental governance methods is analyzed, and the use of economic methods to improve the environment is relatively efficient [22]. The development and limitations of environmental expenditure and revenue policies in the US, the UK, and Sweden are discussed. Moreover, the policy effects of emission reduction in the expenditure and revenue system is explained [23].

Based on the methodologies of previous studies, the Black–Scholes pricing model is an option pricing model of a binary tree, which has great influence on the development of a financial asset pricing model. We studied the impact of environmental pollution liability insurance pricing on the solvency of insurance companies in China, based on the Black–Scholes pricing model. The results of this study could contribute to innovations in China’s carbon financial markets.

### 1.2. Research Design

The use of various economic measures to encourage environmental protection is proposed, including sewage permission and environmental expenditures and subsidies [24]. Using environmental expenditures instead of other types of expenditures can improve expenditure and revenue efficiency while protecting the environment; this proves the existence of a “double dividend” [25]. The collection of carbon expenditure could play a role in reducing carbon emissions so that the overall effect of the expenditure and revenue burden can be reduced. However, some scholars [26] have doubts about the double dividend. The computable general equilibrium (CGE) model was used to analyze the double dividend and it was found that it was not realized [27]. The double dividend of environmental expenditure has two types: strong and weak double dividends. The weak double dividend can protect the environment while reducing the expenditure and revenue burden, whereas the strong double dividend can also increase the efficiency of the expenditure system [28]. For small economies, the double dividend effect on environmental expenditure exists in the long run. Carbon expenditure through the dynamic CGE model was analyzed and there was a weak double dividend of carbon expenditure, but it was difficult to determine the existence of a strong double dividend [29]. Although environmental expenditure leads to an increase in resource prices in the short term, it promotes growth in employment and enterprise innovation in the long term [30].

Ackermana et al. measured the hidden trade-embodied carbon emissions in US–Japan trade, and using the input–output model, he estimated that in 1995, the US’s trade-embodied carbon emissions in US–Japan trade reduced by 14.6 million tons. However, Japan’s carbon emission only increased by 6.7 million tons, and the US–Japan trade saved the world about 7.9 million tons of carbon emissions. This shows that the carbon emissions of the US exceed that of Japan, and reasonable trade between them is conducive to promoting energy-saving and reducing emission [31]. Some scholars [32] hold that carbon emissions reduction expenditure violates the World Trade Organization (WTO) rules. Syunkova (2007) believed that the US carbon emission reduction policy requires importers to purchase emission allowances, which increases the cost of imported products [33]. The life cycle assessment (LCA) method is used to measure hidden carbon emission in the Sino-US trade and estimated China’s carbon emission coefficient based on the US’s carbon emission coefficient. It was found that the US is responsible for 7% to 14% of China’s carbon emission [34].

The carbon emission reduction expenditure meets the General Agreement on Tariffs and Trade (GATT) exception clause, and the implementation of carbon emission reduction expenditure policies affects economic trade [35]. Yan Dong and John Whalley [36] constructed a CGE static model that includes the US, the EU, China, and other regions. They simulated the scenario of the EU and the US levying a tax on imports to reduce carbon emissions, and they found that the carbon emission reduction expenditure has an insignificant impact on global carbon emission and trade. Although it was intended to reduce the imports of the countries that were levied, such as China, their imports increased while their exports decreased. The carbon emission reduction policy under the EU emission system could effectively solve “carbon leakage”. However, the EU’s imposition of a levy to reduce carbon emissions will positively affect its welfare, and the welfare of the countries that are levied will be negatively affected [37]. The impact of carbon emission reduction expenditure on the world trade pattern was studied, and the carbon emission reduction expenditure imposed on China by the EU and the US will inevitably lead to the contraction of China’s export [36]. Ben Lockwood and John Whalley believed that countries upon whom the emission reduction levy is imposed can find other ways to replace it [38].

The above studies provided reference experience and new ideas for this research and related topics. Firstly, this study analyzes the problems existing in the compulsory liability insurance for environmental pollution in China. It proceeds with analyzing the price of compulsory environmental pollution liability insurance using the Black–Scholes pricing model. Moreover, it performs a multivariate regression analysis using the asset and liability data, taken from the annual report, to identify three key factors affecting the solvency adequacy ratio, namely, asset–liability ratio, reflecting the company asset structure; net interest rate, reflecting the asset scale with actual solvency; and loss ratio, reflecting the business quality. Based on the results of regression analysis for the China Insurance Clauses (CIC) company, People’s Insurance Company of China (PICC), and Asia-Pacific Property & Casualty Insurance (API) company, total asset, total debt, capital debt ratio, claim ratio, and net interest rate on assets, have effect on the solvency adequacy ratio.

### 1.3. Incremental Contributions

The use of expenditure and revenue could balance the environmental pollution problems caused by private sector emissions, and the use of taxation and subsidies could promote social and private costs [39]. The government can increase the sewage charging standards for enterprises to reduce pollutant emissions for economic benefits [40]. Due to the difficulty in implementing sewage charges at the enterprise level and the rise in supervision costs, the information asymmetry between government and enterprise supervision has reduced the efficiency of air pollution control [41]. For the long-term development of an enterprise, sewage charges enhance the enterprise’s value and corporate image to achieve good competitiveness, and it does not affect its economic growth negatively [42]. Based on China’s development status, if carbon emissions are included in the tax collection scope in the expenditure and revenue system, it will enhance the harmonious development of the society and distribution system [43]. Halkos (2013) selected the emission reduction data of more than 70 countries from 1980 to 2000 for empirical research, and the results showed that the governments’ capital investment and sulfur dioxide emissions have a negative relationship. However, the level of fiscal expenditure does not affect the amount of carbon emissions [44].

The multi-regional and multi-sector CGE model was used to study the reform effect of EPT and the current EPT method was obtained that is used to mitigate the reduction in domestic GDP and reduce emissions to the minimum [45]. The endogenous growth theory was used to examine the impact of green expenditure reform on the growth of the economy by the authors of [46]. Their experiments showed that through redistribution of inputs, environmental expenditure reform leads to positive growth dividends. A fully differentiated expenditure rate system is more effective than current policies with similar equity performance [47]. The fully differentiated expenditure rate system will increase the overall health benefits of the current partially differentiated expenditure rate system by 43.1%. This shows that differentiation is essential in the design of the pollution expenditure rate system in China. Xinghua Fan et al. (2019) showed the evolution of economic growth, pollution intensity, and resource intensity when collecting environmental expenditure from the perspective of the power system. Empirical research shows that environmental protection expenditure plays an active role in the development of the green economy, which not only promotes economic growth but also saves resources and reduces pollution [48].

Based on the results of previous studies [49,50], the rest of this paper is structured as follows: the second part is the Methods and Materials. It contains the Black–Scholes (B-S) differential equation, multiple regression model, and data collection. The third part is the Results and Discussion. It contains a calculation of the premium rate of the B-S pricing model of environmental liability insurance, and the main effect test on the impact of total asset, total debt, capital debt ratio, claim ratio, and net interest rate on assets on the solvency of insurance company. The fourth part is the Conclusion and Recommendations.

## 2. Methods and Materials

### 2.1. Black–Scholes (B-S) Differential Equation

Since researchers assume that the market price S of financial assets follows geometric Brownian motion [16], there are:(1)dS=μSΔt+σSΔz

Suppose f (S, t) is a function that depends on the market price S and time t, which can be obtained from Formula (2):(2)$$df=(\frac{\partial f}{\partial S}\mu S+\frac{\partial f}{\partial t}+\frac{1}{2}\frac{\partial^{2}f}{\partial S^{2}}\sigma^{2}S^{2})\Delta T+\frac{\partial f}{\partial S}\sigma Sdz$$

During a short time Δt:(3)ΔS=μSΔt+σSΔz
(4)$$\Delta f=(\frac{\partial f}{\partial S}\mu S+\frac{\partial f}{\partial t}+\frac{1}{2}\frac{\partial^{2}f}{\partial S^{2}}\sigma^{2}S^{2})\Delta t+\frac{\partial f}{\partial S}\sigma S\Delta z$$

It can be seen from Formulas (3) and (4) that there is a common term Δz, so as long as the combination of the appropriate derivative financial assets and the underlying assets is selected, the uncertainty can be eliminated. A combination of a short position of unit-derived financial asset and long position of $\frac{\partial f}{\partial S}$ unit underlying asset can be constructed. The value of the asset portfolio is represented by π:(5)$$\Pi =-f+\frac{\partial f}{\partial S}S$$

After the time Δt, the value change Δπ is:(6)$$\Delta \Pi =-\Delta f+\frac{\partial f}{\partial S}\Delta S$$

The following can be obtained by the simultaneous solution of Formulas (4)–(6):(7)$$\Delta \Pi =(-\frac{\partial f}{\partial t}-\frac{1}{2}\frac{\partial^{2}f}{\partial S^{2}}\sigma^{2}S^{2})\Delta t$$

It can be seen from Formula (7) that the value of the portfolio is risk-free after the time Δt. Because the formula does not contain the term Δz, the instantaneous yield of the portfolio is equal to the risk-free return rate. Therefore, under the condition without arbitrage:(8)ΔΠ=rΠΔt

It can be obtained by the simultaneous solution of Formulas (6), (7), and (8) that:(9)$$(\frac{\partial f}{\partial t}+\frac{1}{2}\frac{\partial^{2}f}{\partial S^{2}}\sigma^{2}S^{2})\Delta t=r(f-\frac{\partial f}{\partial S}S)\Delta t$$

The following can be obtained by simplifying and organizing:(10)$$\frac{\partial f}{\partial t}+rS\frac{\partial f}{\partial S}+\frac{1}{2}\sigma^{2}S^{2}+\frac{\partial^{2}f}{\partial S^{2}}=rf$$

This is the Black–Scholes differential equation, which applies to the pricing of all derivative assets whose price depends on the price S of the underlying financial assets. When the call option expires, the option value is CT = max (ST − K, 0). Therefore, let C (S, t) represent the price of the European call option at the time t, then:(11)$$\begin{array}{c} \frac{\partial C}{\partial t}+rS\frac{\partial C}{\partial S}+\frac{1}{2}\sigma^{2}S^{2}\frac{\partial^{2}C}{\partial S^{2}}=rC\\ C_{T}=\max\left(S_{T}-K,0\right) \end{array}$$

The following can be obtained:(12)$$C=SN\left(d_{1}\right)-Ke^{-r\left(T-t\right)}N\left(d_{2}\right)$$
where
(13)$$d_{1}=\frac{\ln(S/K)+(r+\sigma^{2}/2)(T-t)}{\sigma \sqrt{\left(T-t\right)}}$$
(14)$$d_{2}=d_{1}-\sigma \sqrt{\left(T-t\right)}$$
(15)$$\sigma =\sqrt{Var\left(dS/S\right)}$$
where C is the price of call option at time t, S is the market price of the underlying asset at time t; K is the execution price of an option contract; r is the annual rate to represent the risk-free interest rate; T is the term of an option contract, usually expressed in years; and N (x) is the cumulative probability distribution function of standard normal distribution. It can be seen that when the call option expires: *C_T_* = max (*K* − *ST*, 0), then the option value is:(16)$$\begin{array}{c} \frac{\partial C}{\partial t}+rS\frac{\partial C}{\partial S}+\frac{1}{2}\sigma^{2}S^{2}\frac{\partial^{2}C}{\partial S^{2}}=rC\\ C_{T}=\max\left(K-S_{T},0\right) \end{array}$$

Because the normal distribution has the numerical characteristics of N (x) = 1 − N (x), the pricing formula of the put option is:(17)$$P=Ke^{-r(T-t)}N(-d_{2})-SN(-d_{1})$$

B-S is a pricing model of European options. The analysis of the characteristics of compulsory liability insurance corresponds to the European put option; Equation (17) presents the pricing formula. Then, the insurance premium rate is the insurance premium divided by the insurance amount, as in Equation (18):(18)$$\bar{p}=e^{-r(T-t)}N(-d_{2})-\frac{S}{K}N(-d_{1})$$
where P is the premium, S is the market price of the underlying asset at time t; K is the underwriting amount; r is the annual rate to represent the risk-free interest rate; T is the term of the option contract, usually expressed in years; and N (x) is the cumulative probability distribution function of the standard normal distribution.

### 2.2. Multiple Regression

The multiple regression method is used for the analysis of the above-discussed data. First, the data taken from the annual report is used to calculate three factors that affect the solvency adequacy ratio. In this paper, the asset liability ratio that reflects the company’s asset structure, the asset net interest rate that reflects the actual solvency, and the loss ratio that reflects the business quality are being used. The maximum likelihood estimation and the ordinary least-squares evaluating model was applied to examine the main effects between independent variables and dependent variables [51,52]. Suppose the regression estimation expression is as follows:(19)$$SAR_{it}=\alpha_{0}+\alpha_{1}TA_{it}+\alpha_{2}TD_{it}+\alpha_{3}CDR_{it}+\alpha_{4}CR_{it}+\alpha_{5}NIRA_{it}+e_{it}$$
where *i* denotes the *i*th respondent; solvency adequacy ratio (*SAR*) is the dependent variable in this study. *TA*, *TD*, *CDR*, *CR*, and *NIRA* represent total asset, total debt, capital debt ratio, claim ratio, and net interest rate on assets, respectively.

### 2.3. Data Collection

#### 2.3.1. B-S Calculation of Pricing Model

(1) Loss Calculation

Based on the premium income of environmental liability insurance, the premium income of liability insurance, and the amount of liability insurance compensation, the loss in compensation of environmental liability insurance is roughly calculated [53]. It calculates the compensation amount of environmental liability insurance over the years. The premium income and compensation of liability insurance in 2007 can be derived in Table 1 as follows.

According to China’s compulsory liability insurance network for environmental pollution, the national environmental liability insurance premium income in 2016 was CNY 280 million (Data Source: China Environmental Pollution Liability Insurance Network: http://www.chinaepli.com/ (accessed on 31 December 2017), with over 4000 insurance companies. According to the statistics from the National Development and Reform Commission, by the first half of 2017, the total number of enterprises reached to 34,339 (Data Source: China Reporting Network: http://data.chinabaogao.com/ (accessed on 31 December 2017). Here is a rough calculation of the loss of environmental liability insurance. Accordingly, environmental liability insurance compensation/liability insurance compensation = environmental liability insurance premiums/liability insurance premiums * liability insurance compensation/liability insurance premiums. The amount of environmental liability insurance compensation for the past years is presented in Table 2, which is calculated based on Table 1.

Accordingly, loss amount = total number of companies insured for environmental liability insurance/total number of companies * environmental liability insurance compensation. Therefore, loss amount = 4000/34,339 * environmental liability insurance compensation. The amount of loss over the years is presented in Table 3, which is calculated based on Table 1.

Because the B-S model assumes that there is no arbitrage in the market, to obtain more rigorous data, the data of 2016 are taken as the benchmark for Table 3, and the data of the remaining years are revised with an inflation rate of 4% to obtain Table 4, which is calculated based on Table 1.

(2) Establish Deductibles and Compensation Limits

China began piloting the compulsory environmental pollution liability insurance in 2013, and the scale of environmental liability insurance has been expanding since then. Therefore, to obtain more rigorous data, the underwriting amount of 2.1703 million was selected as the average loss amount for the period 2013–2016, which is more in line with the actual development of environmental liability insurance [54]. This paper selects the round figure of CNY 2.2 million as the compensation limit. According to the *Journal of Safety and Environment,* published every two months for the past ten years, it is evident that most of the environmental pollution losses are more than CNY 100 thousand, which is thus set as the absolute deductible.

#### 2.3.2. Pricing of B-S Model of Environmental Liability Insurance

As a kind of liability insurance, environmental liability insurance has no clear insurance subject, so it can only roughly express the current market price with the amount of loss compensation. It performs a hypotheses testing where SPSS software is used to perform a single-sample K-S test on the loss amount of environmental liability insurance. The data in Table 4 are used to obtain the results presented in Table 5, and obtain the *p*-value: *p* = 0.2 > 0.05, which indicates that the current market price of environmental liability insurance follows the normal distribution, i.e., follows the geometric Brownian motion.

At the same time, from Figure 1 it is also evident that the distribution of the line chart of the loss amount coincides with the left half of the normal distribution chart.

### 2.4. Data Reduction by Different Companies

In this section, data from the macro- and micro-levels of the liability insurance category to which the contract energy management insurance belongs are collected, organized, and analyzed. On the macro-level, the data are analyzed from a nationwide perspective. On the micro-level, three large insurance companies that can provide contract energy management insurance are selected and relevant data are collected from their annual reports for analysis.

#### 2.4.1. Macro-Level

(1) Types of responsible property: according to the detailed list of classifications of liability insurance products published in 2013, it can be seen that there are nine general categories of responsible property, with a total of 55 types. The contract energy management insurance products discussed in this article, that is, income liability insurance for energy-saving projects, should fall into the category of product liability insurance. It is worth mentioning that, due to the large number of product categories, only the top 20 are listed by the potential market Forecast of Energy Performance Contracting (EPC) industry [55].

(2) Premium and compensation for national liability insurance. This article has used the 10-year data collected from the National Bureau of Statistics for the period 2007–2016. The data will be arranged in the form of graphs and linear graphs (as shown in Figure 2 and Figure 3).

#### 2.4.2. Micro-Level

In this section, three large insurance companies that provide liability insurance for energy-saving projects are selected and sorted out according to the statistics published in the annual reports for the period 2012–2017. The data include the company’s actual and minimum capital, solvency adequacy ratio, and the ranking of liability insurance in the company’s business [56]. The data in this paragraph are from 2012–2017 (the data in Asia-Pacific Property & Casualty Insurance Co., Ltd. (API), Shenzhen, China, is from 2010–2016).

(1) China Insurance Clauses (CIC) Company

CIC is the second insurance company with legal personality in China. In 2013, the company cooperatively developed the energy-saving income liability insurance. The actual capital and minimum capital of CIC is described in Figure 4.

It should be noted, that due to the implementation of the second generation of compensation regulation in 2016, the measurement methods of actual capital are different, so there will be a greater improvement in the data. Based on this, the solvency of the insurance company for debts can be calculated by adequacy ratio = actual capital divided by minimum capital, as shown in Table 6.

According to the calculated solvency adequacy ratio, it can be seen that the company’s solvency has reached the level of sufficient class II in these five years. From the statistics provided in the company’s annual report, it is evident that the change in the solvency rate is mainly because of the change in the company’s capital structure and business. In the 2012 annual report, we learned that the actual capital in 2011 was CNY –9962.29 million, which directly led to the decrease of solvency adequacy rate to −346%, which can also explain this problem. The following data shown in Table 7 are obtained and calculated from the annual report of the company:

(2) People’s Insurance Company of China (PICC)

PICC is a comprehensive insurance company, which is one of the largest insurance companies in China. In 2016, the company signed and developed ESCO service quality assurance insurance and EMC contract future income right insurance with API. The actual capital and minimum capital of PICC are described in Figure 5.

The calculation of solvency adequacy ratio is as follows shown in Table 8:

In addition to the above, the annual disclosure report of PICC also published the ranking of the main types of insurance in the company’s business. Overview of liability insurance of PICC are described in Figure 6.

According to the statistics from the annual report, the retained premium of PICC increases to a certain extent every year, which has led to the growth of minimum capital. The rapid growth of business and the maintenance of higher profitability also affect the minimum capital and actual capital at the same time. The decrease of solvency adequacy ratio in 2015 is due to the fact that although the company redeemed the subordinated term debt in full and completed financing supplementary capital, the growth rate of actual capital is lower than the growth rate of minimum capital, which also shows that a change affecting the growth rate of both will also affect the change of adequacy ratio. In addition, from the data of liability insurance, liability insurance has always been steadily ranked in the top three, indicating that there will be more personnel assigned by the company, which is also enough to show its importance. The following data are obtained and calculated from the annual report of the company, as shown in Table 9.

(3) Asia-Pacific Property & Casualty Insurance Company (API)

API is a company mainly engaged in property loss insurance, liability insurance, credit insurance, and guarantee insurance. In 2016, the company jointly developed ESCO service quality assurance insurance and EMC contract future earning right insurance with PICC. Actual and minimum capital of API are described in Figure 7.

The calculation of solvency adequacy ratio based on the capital data is as follows shown in Table 10:

The summary of liability insurance according to the annual report is as follows shown in Figure 8:

Compared to the previously discussed two large companies, the scale of API is relatively small, and the establishment time is not long. Therefore, there exist large fluctuations in the statistics provided in the data. This is mainly because the premium income from financing and business of small companies is very unstable. For example, the solvency adequacy ratio has shown an increase of increased 338% in 2012 because the company received CNY 1.01 billion of financing. However, it decreased by 148% in 2013 because the comprehensive income was CNY −100.93 million, which affected the actual capital. Although the data on premium income in 2015 exhibited negative growth, the adequacy ratio rose by 8.07% because the decline in the actual capital was lower than the minimum capital decline. Despite the implementation of the second-generation compensation, the fact that the adequacy ratio exceeded 500% in 2016 is also because of an increase of CNY 2 billion of financing. Even though breaking through 500% is good news, it is not necessarily a long-term and stable result for small and medium-sized companies. Information on API is shown in Table 11 as follows:

## 3. Results and Discussion

### 3.1. Calculation of Premium Rate of B-S Pricing Model of Environmental Liability Insurance

#### 3.1.1. The First Step Is to Select a Risk-Free Interest Rate

The higher the risk-free interest rate of the European put option, the greater the opportunity cost will be, and the higher will be the positive correlation. Under the premise of satisfying all the assumptions of the B-S pricing model, the risk-free interest rate is selected. The risk-free interest rate is usually the average of one-year treasury bonds in the most recent year. That is, the average value of the interest rate of one-year national debt in 2017 is selected.

#### 3.1.2. The Second Step Is to Calculate the Volatility

The possibility of the option buyer’s profitability is consistent with the change in the price of the insurance subject. The greater the price volatility of the insurance subject, the greater the possibility of the option buyer’s profitability, and the opposite is true for the seller. The standard deviation of the compound yield within a certain period is the volatility of the price of the subject-matter insured. u(i) is used to represent the price volatility of the insurance subject in the i-th period, and it can be calculated from the data in Table 4.
(20)$$E(u_{i})=1.1799;{\left[E(u_{i})\right]}^{2}=1.3921;\;E(u_{i}^{2})=1.3954$$

Then, the Formula (21) is obtained:(21)$$\sigma =\sqrt{\frac{E{\left(u_{i}-E(u_{i})\right)}^{2}}{T-t}}$$

Therefore, σ = 0.0574.

#### 3.1.3. The Third Step Is to Calculate the Insurance Premium Rate

The market price of the insurance subject is equal to the difference between the insurance underwriting amount PV(S) and the policy’s expected loss P(S). Because of the loss uncertainty, the comparison between the average loss of CNY 1.3775 million in the past years (which can be calculated based on the data in Table 4) and the average number of environmental pollution accidents (106 times) that happened in the past (which can be obtained from the data in Table 3) can be made to obtain the expected loss of the policy.
(22)$$S=PV(S)-P(S)=\frac{K-P(t)}{1+r}=\frac{220-1.2995}{1+2.71}=212.9296$$

The following can be calculated by substituting the above results into Formula (14):(23)$$d_{1}=\frac{\ln(S/K)+r+\sigma^{2}/2}{\sigma }=-0.0733$$
(24)$$d_{2}=d_{1}-\sigma \sqrt{T-t}=-0.0733-0.0559=-0.1292$$

Based on the table of probability distribution, we can obtain:(25)$$N(d_{1})=0.5292,N(-d_{2})=10.5514$$

Therefore, the insurance premium rate should be
(26)$$\bar{p}=e^{-r(T-t)}N(-d_{2})-\frac{S}{K}N(-d_{1})=2.44$$

Based on the calculations above of the B-S pricing model, the premium income of environmental liability insurance, the amount of liability insurance compensation, and the loss in compensation of environmental liability insurance are roughly calculated for the CIC, PICC and API, which is used for the main effect test of the regression analysis.

### 3.2. Main Effect Test

#### 3.2.1. CIC

Taking the original data in Table 6 and Table 7 of CIC as an example, the econometric software Stata 16 is used to perform regression based on the confidence setting of 90%, and the following output tables are obtained after processing the data of CIC as presented in Table 12 and Table 13.
(27)$$SAR_{i}=-3.2878+0.2929TA_{i}+0.0449CDR_{i}+1.8823CR_{i}-0.1744NIRA_{i}$$

From the above regression equation, it is evident that, among all factors, total asset, claim ratio, and the net interest rate on assets have a greater influence on the solvency adequacy ratio. Based on the higher R-squared, it can be seen that these factors have only a certain degree of impact on the solvency adequacy ratio. Furthermore, it can be seen that the factor capital debt ratio has only a certain degree of impact on the solvency adequacy ratio but not a decisive one.

#### 3.2.2. PICC

Taking the original data of PICC as an example, the original data are presented in Table 8 and Table 9. Following a similar procedure, the following results are obtained after processing the data of PICC. Stata 16 is used to perform regression based on the confidence setting of 90%, and the following two output tables are obtained:

In the regression statistics presented in Table 14 and Table 15, it is evident that independent variables and the solvency adequacy ratio are also highly correlated, and the fitting degree is good from the point of standard error. Furthermore, the significance F of the analysis of variance table (ANOVA table) is 0.1471, which also shows that the significance of the regression is not high due to the insufficient amount of data. However, from the final regression parameter table, the equation is as follows:(28)$$SAR_{i}=-12.7481+9.0461TA_{i}-0.0001TD_{i}+19.6514CDR_{i}+8.9803CR_{i}-56.4684NIRA_{i}$$

It shows that total asset, total debt, capital debt ratio, claim ratio, and net interest rate on assets have a great influence on the solvency adequacy ratio. However, the *p*-value is higher than the significance level of 0.05; it still can be seen that these factors have only a certain degree of impact on the solvency adequacy but not a decisive one.

#### 3.2.3. API

Taking the data of API as an example, the original data are presented in Table 10 and Table 11. Following a similar procedure, the following results are obtained after processing the data of API. Stata 16 is used to perform regression based on the confidence setting of 90%, and the following two output Table 16 and Table 17 are obtained:

In the regression table, a multiple R (correlation coefficient) of 0.9784, which is used to judge the correlation degree between independent variables and the solvency adequacy ratio, indicates that the relationship between them is highly correlated. The adjusted R square of 0.7744 also shows that the independent variable can account for 77.44% of the dependent variable, and the remaining 22.56% should be explained by other factors. The standard error is 0.7743, which indicates that the fitting degree is good. In the ANOVA table, the regression effect is mainly judged by F-statistics. Since the data used for analysis are for only 7 years (2010 to 2016), the sample size is small; the significance of F-statistics in the table reaches 0.3431, indicating that the regression effect is not very significant. The *p*-value also shows that all coefficient values are greater than the significance level of 0.05, indicating that there is a certain correlation between the independent variable and the dependent variable, but the correlation is not large. However, the regression equation obtained from coefficients is:(29)$$SAR_{i}=-62.5330+1.6856TA_{i}-2.2842TD_{i}+91.1215CDR_{i}+0.1527CR_{i}-90.3570NIRA_{i}$$

### 3.3. Robustness Test

The regression coefficients can be unstable for various reasons. Regression analysis requires corporate performance to be normally distributed and sensitive to outliers. Outlier problems, collinearity problems, and heteroscedasticity problems can all lead to biased regression results. Additionally, we cannot understand the changing process of the influence trend of total asset, total debt, capital debt ratio, claim ratio, and net interest rate on assets, on the solvency of the insurance companies through regression analysis, whereas quantile regression can solve this problem very well [57]. The regression results are shown in Table 18.

Moreover, we will address the robustness test. Here, we mainly replace the initial regression with the quantile regression method to obtain more effective estimation results. This paper uses the results of quantile number as the regression coefficient, and the regression results are described in Figure 9.

Three robustness tests are carried out in this section, and the initial results are passed: from Table 18, in terms of quantiles, insurance companies in different positions are affected by TA, TD, CDR, CR, and NIRA to different degrees with the fluctuations of the estimators in the confidence interval of Figure 9, but the results are all significant.

## 4. Conclusions and Recommendations

### 4.1. Conclusions

According to the data analysis in the previous section, the following results are obtained. 

(1) Based on the B-S pricing model, the premium rate of compulsory environmental liability insurance is 2.44%, which is higher than other liability insurance. This is an important factor restricting the market share of environmental liability insurance. The insurance premium rate of environmental liability insurance needs to be further optimized. 

(2) In this paper, the premium and loss ratio of 10-year liability insurance is selected, and the relevant data obtained by averaging the proportion of environmental liability insurance premium to liability insurance in 2016 may be slightly different from the actual situation, but the development trend of compulsory environmental liability insurance is generally unchanged. Moreover, it performs a multivariate regression analysis using the asset and liability data, taken from the annual report, to identify three key factors affecting the solvency adequacy ratio, namely, capital debt ratio, reflecting the company asset structure; net interest rate on assets, reflecting the asset scale with actual solvency; and claim ratio, reflecting the business quality.

(3) Based on the results of regression analysis and robustness test for the China Insurance Clauses (CIC) company, People’s Insurance Company of China (PICC), and Asia-Pacific Property & Casualty Insurance (API) company, it is shown that the effect of total asset, total debt, capital debt ratio, claim ratio, and net interest rate on assets on the solvency adequacy ratio is significant, with respect to the size of the coefficients.

### 4.2. Recommendations

According to the analysis of the annual report data of three companies, it can be seen that the companies that provide liability insurance for energy-saving projects now meet the requirements of “solvency adequacy ratio of more than 100%” stipulated by the China Insurance Regulatory Commission. When the company’s environment is stable, the stability of contract energy management insurance is greatly guaranteed. At the macro-level, the premium and compensation of China’s liability insurance can keep a balance of 2:1 to allow a certain extent of error, even if the growth rate of compensation is slightly higher than that of premium. In the long run, the balance is difficult to break, as long as there are no extreme events. Although it is not very difficult for large companies with stable financing and business sources to maintain solvency adequacy ratio at about 200%, for small and medium-sized companies, the uncertainty of financing and business sources is one of the reasons for the difficulty in ensuring adequacy. Therefore, such companies often choose to cooperate with large companies to develop products, not only to ensure financing but also to attract some business through cooperation. Whether it is a large company or a general small or medium-sized company, liability insurance is kept stable in the front position. It is necessary for social development, including energy-saving income liability insurance. The stability of insurance products depends on the company’s stable structure. Based on the regression analysis of the data, it can be seen that the capital debt ratio and the net capital interest rate have a certain effect on solvency, and both of them represent the asset structure of the company, including the stability of financing and business. Stable financing and business income can ensure the stable growth of actual capital and also helps in maintaining the minimum capital requirement. The main change of the second-generation compensation implemented in 2016 is to recognize the difference between liabilities, and the liability side reduced the requirements for insurance contract reserves, resulting in the actual capital under the second-generation caliber being higher than that under the previous first-generation caliber. Finally, the solvency adequacy ratio of the second generation is often higher than the solvency adequacy ratio of the first generation [56]. Overall, the existing energy-saving liability insurance in China has a certain level of stability. With the increasing premium rate of compulsory environmental liability insurance in China, it is necessary for insurance companies to increase total asset, capital debt ratio, and claim ratio, and reduce total debt and net interest rate on assets, in order to increase solvency.

## Figures and Tables

**Figure 1 ijerph-20-01630-f001:**
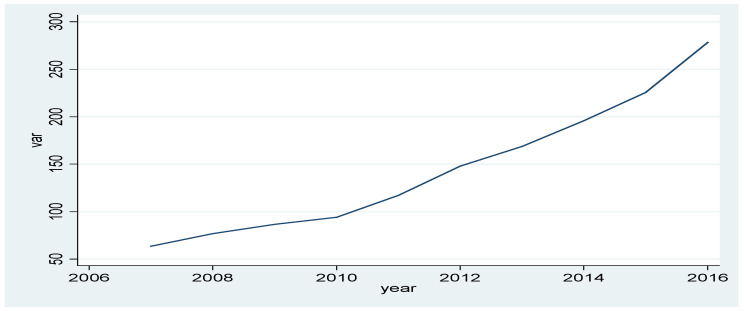
Statistics of the amount of environmental pollution losses in 2007–2016. Data source: Table 4.

**Figure 2 ijerph-20-01630-f002:**
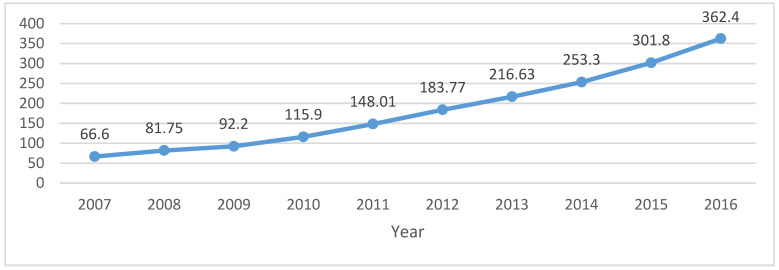
Premium of liability insurance of property insurance company (2007–2016, CNY 100 million). Data source: China National Bureau of Statistics.

**Figure 3 ijerph-20-01630-f003:**
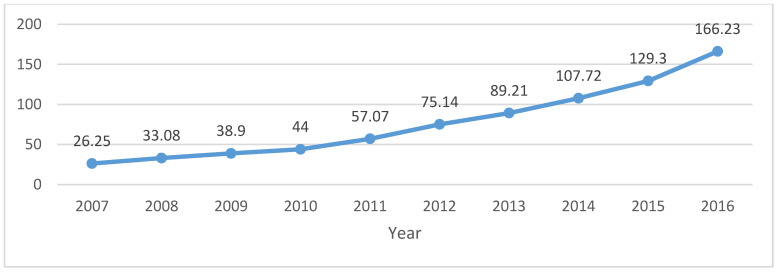
Compensation of liability insurance of property insurance company (2007–2016, CNY 100 million). Data source: China National Bureau of Statistics.

**Figure 4 ijerph-20-01630-f004:**
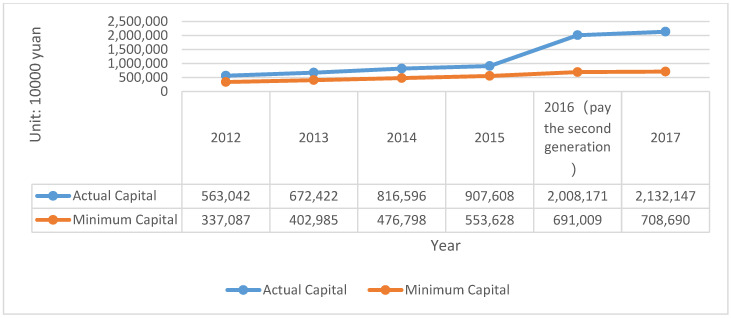
The actual capital and minimum capital of CIC (2012–2016). Source: Annual report of CIC.

**Figure 5 ijerph-20-01630-f005:**
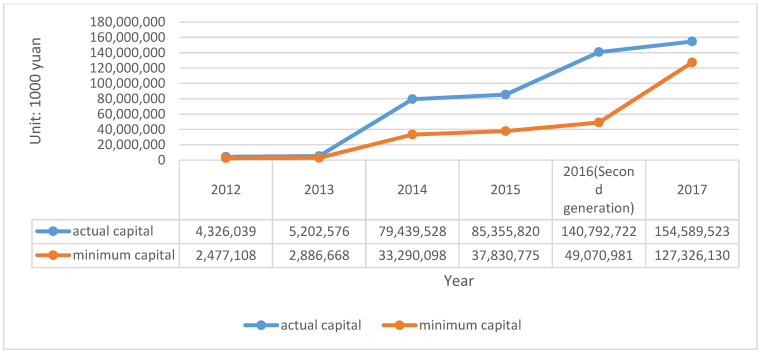
The Actual Capital and Minimum Capital of PICC (2012–2017). Data source: Annual report of PICC.

**Figure 6 ijerph-20-01630-f006:**
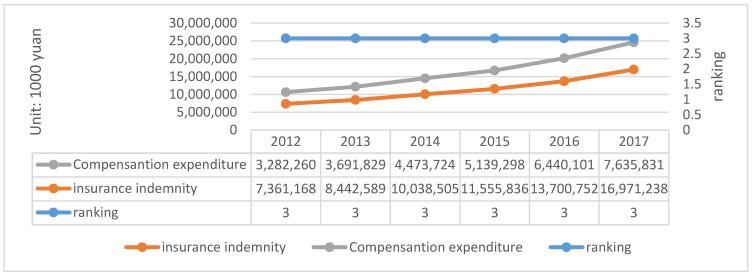
Overview of liability insurance of PICC (2012–2017). Data source: Annual report of PICC.

**Figure 7 ijerph-20-01630-f007:**
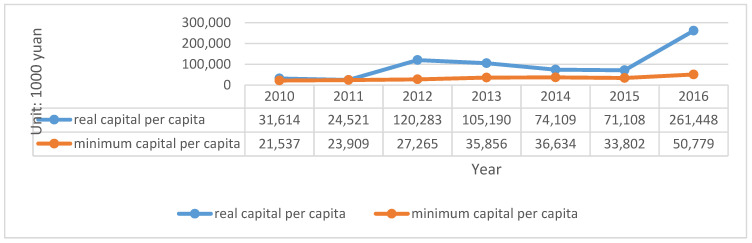
Actual and minimum capital of API (2010–2016). Source: Annual report of API.

**Figure 8 ijerph-20-01630-f008:**
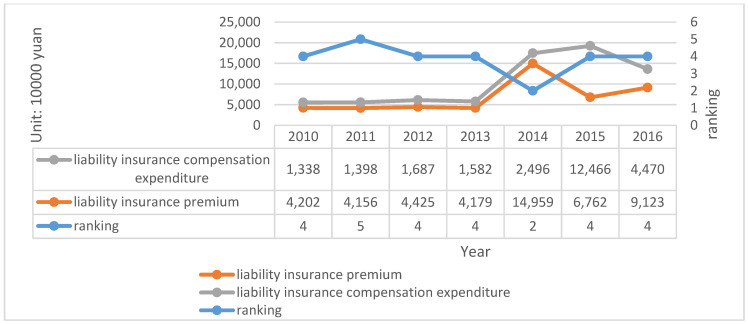
Overview of API liability insurance (2010–2016). Source: Annual report of API.

**Figure 9 ijerph-20-01630-f009:**
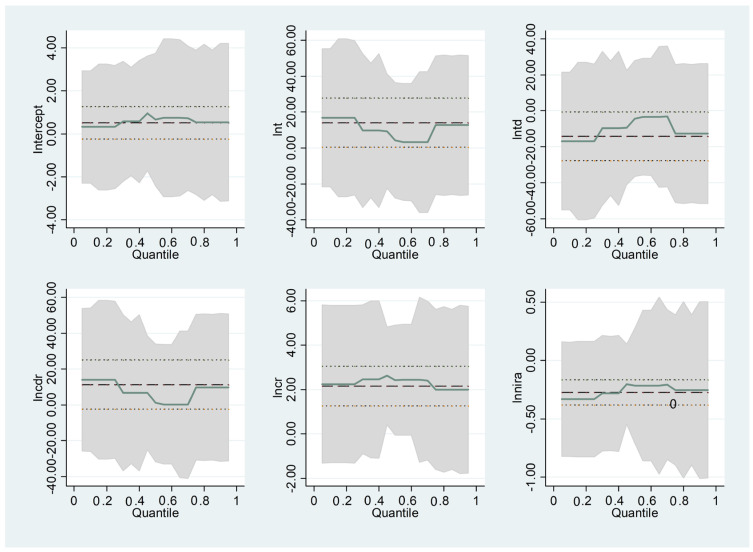
Robustness test for bootstrap and cluster robust standard errors.

**Table 1 ijerph-20-01630-t001:** Statistics of premium income and compensation of liability insurance in 2007.

Unit: CNY 100 Million	2007	2008	2009	2010	2011	2012	2013	2014	2015	2016
Liability insurance Premium	66.6	81.75	92.2	115.9	148.01	183.77	216.63	253.3	301.8	362.4
Liability insurance compensate	26.25	33.08	38.9	44	57.07	75.14	89.21	107.72	129.3	166.23

Data source: China Environmental Pollution Liability Insurance Network: http://www.chinaepli.com/ (accessed on 31 December 2017); China Reporting Network: http://data.chinabaogao.com/ (accessed on 31 December 2017).

**Table 2 ijerph-20-01630-t002:** Statistics of the amount of environmental liability insurance compensation in 2007–2016.

Unit: CNY 10,000	2007	2008	2009	2010	2011	2012	2013	2014	2015	2016
Liability insurance compensate	2529.6	503.39	591.96	669.57	868.46	1143.4	1357.5	1639.2	1967.6	2529.6

**Table 3 ijerph-20-01630-t003:** Statistics of the loss amount of enterprises having environmental accidents during 2007–2016.

Unit: CNY 10,000	2007	2008	2009	2010	2011	2012	2013	2014	2015	2016
Liability insurance premium	43.94	55.37	65.12	73.65	95.53	125.78	149.33	180.31	216.44	278.25

**Table 4 ijerph-20-01630-t004:** Statistics of the loss amount of enterprises having environmental accidents excluding inflation from 2007 to 2016.

Unit: CNY 10,000	2007	2008	2009	2010	2011	2012	2013	2014	2015	2016
Liability insurance premium	63.45	76.76	86.65	94.09	117.16	148.09	168.78	195.65	225.46	278.25

**Table 5 ijerph-20-01630-t005:** One-Sample Kolmogorov–Smirnov test form for the loss amount in 2007–2016.

		VAR00001
N		10
Normal Parameters ^a,b^	Mean	145.4340
	Std. Deviation	71.06075
Most Extreme Differences	Absolute	0.165
	Positive	0.165
	Negative	−0.124
Test Statistic		0.165
Asymp. Sig. (2-tailed)		0.200

^a^. Test distribution is normal. ^b^. Calculated from data.

**Table 6 ijerph-20-01630-t006:** The solvency adequacy ratio of CIC (2012–2017).

Year	2012	2013	2014	2015	2016	2017
Solvency adequacy ratio	167%	166.86%	171.27%	163.94%	290.61%	300.86%

Data source: Annual report of CIC.

**Table 7 ijerph-20-01630-t007:** Information of CIC (2012–2017).

Unit: CNY 1000	2012	2013	2014	2015	2016	2017
Total Asset	32,774,662	37,920,228	42,503,286	53,034,399	59,756,316	69,210,296
Total Debt	25,785,128	29,968,904	32,035,899	39,880,522	46,229,646	54,550,658
Capital debt ratio	78.67%	79.03%	75%	75.19%	77.36%	78.81%
Claim ratio	38.83%	36.87%	39.25%	35.48%	40.31%	43.66%
Net interest rate on assets	6.66%	3%	4.69%	4.61%	1.47%	1.86%

Data source: Annual report of CIC.

**Table 8 ijerph-20-01630-t008:** Solvency adequacy ratio of PICC (2012–2017).

Year	2012	2013	2014	2015	2016	2017
Solvency adequacy ratio	175%	180%	239%	226%	287%	278%

Data source: Annual report of PICC.

**Table 9 ijerph-20-01630-t009:** Information of PICC (2012–2017).

Unit: CNY 1000	2012	2013	2014	2015	2016	2017
Total Asset	290,424	319,424	366,130	420,420	475,949	524,566
Total Debt	244,974	261,920	280,355	311,469	356,637	391,452
Capital debt ratio	84.35%	81.99%	76.57%	74.08%	74.93%	74.62%
Claim ratio	44.58%	42.73%	44.56%	44.47%	47%	44.99%
Net interest rate on assets	3.58%	3.30%	4.12%	5.19%	3.78%	3.77%

Data source: Annual report of PICC.

**Table 10 ijerph-20-01630-t010:** Solvency adequacy ratio of API (2010–2016).

Year	2010	2011	2012	2013	2014	2015	2016
Solvency adequacy ratio	146%	102%	441%	293%	202%	210%	514%

Source: Annual report of API.

**Table 11 ijerph-20-01630-t011:** Information on API (2010–2016).

Unit: CNY 1000	2010	2011	2012	2013	2014	2015	2016
Total Asset	198,752	250,982	365,561	364,060	396,572	361,251	545,575
Total Debt	146,200	221,600	234,725	235,325	271,347	246,375	276,022
Capital Debt Ratio	73.55%	88%	64%	64%	68%	68.20%	50.59%
Claim Ratio	31.84%	33.63%	38.12%	37.85%	16.68%	184%	48.99%
Net Interest Rate on Assets	−3.56%	−9.21%	0.09%	0.08%	−2.66%	−2.68%	−8.19%

Source: Annual report of API.

**Table 12 ijerph-20-01630-t012:** Analysis of variance of CIC data.

	Df	SS	MS	F	Significance F
Regression Analysis	4	0.4256	0.1064	10.31	0.2290
Residual	1	0.0103	0.0103		
Total	5	0.4359			

**Table 13 ijerph-20-01630-t013:** Regression parameter table of CIC data.

Variables	SAR ^1^	Standard Error	T
TA ^2^	0.2929 ***	0.0414	7.07
TD ^3^	-	-	-
CDR ^4^	0.0449	1.0892	0.04
CR ^5^	1.8823 ***	0.4320	4.36
NIRA ^6^	−0.1744 ***	0.0372	−4.69
Constant	−3.2878	-	-
R-squared	0.9763		
Adjusted R-squared	0.8816		

^1^ SAR: solvency adequacy ratio; ^2^ TA: total asset; ^3^ TD: total debt; ^4^ CDR: capital debt ratio; ^5^ CR: claim ratio; ^6^ NIRA: net interest rate on assets. *** represents *p* < 1%.

**Table 14 ijerph-20-01630-t014:** Analysis of variance of PICC data.

	Df	SS	MS	F	Significance F
Regression Analysis	5	1.2952	0.2590	26.24367	0.1471
Residual	1	0.0099	0.0099		
Total	6	1.3051			

**Table 15 ijerph-20-01630-t015:** Regression parameter table of PICC Data.

Variables	SAR ^1^	Standard Error	T
TA ^2^	9.0461 ***	2.7400	3.2988
TD ^3^	−0.0001 ***	3.8400	−3.2292
CDR ^4^	19.6514 ***	8.6160	2.2808
CR ^5^	8.9803 *	4.6747	1.9210
NIRA ^6^	−56.4684 ***	15.9622	−3.5376
Constant	−12.7481 *	7.1748	−1.7768
R-squared	0.9924		
Adjusted R-squared	0.9546		

^1^ SAR: solvency adequacy ratio; ^2^ TA: total asset; ^3^ TD: total debt; ^4^ CDR: capital debt ratio; ^5^ CR: claim ratio; ^6^ NIRA: net interest rate on assets. *** and * represent *p* < 1% and *p* < 10%.

**Table 16 ijerph-20-01630-t016:** Analysis of variance table of API data.

	Df	SS	MS	F	Significance F
Regression Analysis	5	13.4697	2.6939	4.4926	0.3431
Residual	1	0.5996	0.5996		
Total	6	14.0693			

**Table 17 ijerph-20-01630-t017:** Regression parameter table of API data.

Variables	SAR ^1^	Standard Error	T
TA ^2^	1.6856 **	8.6000	1.9605
TD ^3^	−2.2843 *	1.1800	−1.9283
CDR ^4^	91.1216 *	52.5091	1.7353
CR ^5^	0.1527	0.6198	0.2464
NIRA ^6^	−90.3570 *	48.4435	1.8652
Constant	−62.5330 *	37.1771	−1.6820
R-squared	0.9574		
Adjusted R-squared	0.7744		

^1^ SAR: solvency adequacy ratio; ^2^ TA: total asset; ^3^ TD: total debt; ^4^ CDR: capital debt ratio; ^5^ CR: claim ratio; ^6^ NIRA: net interest rate on assets. ** and * represent *p* < 5% and *p* < 10%.

**Table 18 ijerph-20-01630-t018:** Robustness test of quantile regression.

	(1)	(2)	(3)	(4)
	SAR ^1^ OLS	SAR ^1^ QR_10	SAR ^1^ QR_50	SAR ^1^ QR_90
TA ^2^	14.22 ** (5.565)	16.87 *** (0.000)	4.358 * (25.94)	12.73 *** (2.273)
TD ^3^	−14.19 ** (5.568)	−16.83 *** (0.000)	−4.313 * (25.96)	−12.70 *** (2.275)
CDR ^4^	11.29 * (5.604)	13.99 *** (0.000)	1.044 * (25.56)	9.795 *** (2.371)
CR ^5^	2.162 *** (0.366)	2.243 *** (3.52)	2.424 * (1.427)	1.997 *** (0.115)
NIRA ^6^	−0.274 *** (0.0442)	−0.332 *** (3.98)	−0.217 * (0.142)	−0.253 *** (0.0163)
Time control	YES	YES	YES	YES
Individual control	YES	YES	YES	YES
_cons	0.524	0.319 ***	0.662	0.539 ***

^1^ SAR: solvency adequacy ratio; ^2^ TA: total asset; ^3^ TD: total debt; ^4^ CDR: capital debt ratio; ^5^ CR: claim ratio; ^6^ NIRA: net interest rate on assets. ***, ** and * represent *p* < 1%, *p* < 5% and *p* < 10%. Standard errors in parentheses.

## Data Availability

Not applicable.

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
