# Peer review of "Environmental Pollution Liability Insurance Pricing and the Solvency of Insurance Companies in China: Based on the Black–Scholes Model"

_ijerph, 2023, doi:10.3390/ijerph20021630_

Round 1

Reviewer 1 Report

Environmental pollution liability insurance is becoming increasingly important for China to achieve its emission reduction targets. Insurance pricing is a crucial factor restricting the market share of environment pollution liability insurance, from the perspective of the Black-Scholes pricing model, in turn has influenced on the solvency of insurance companies in China.

This study analyzes the problems existing in the compulsory liability insurance for environmental pollution in China and performs a multivariate regression analysis. The research questions are important and the conclusions are clear.

In order to meet the requirements of publication, I think the paper needs to be substantially modified in at least the following aspects.

Firstly, the introduction uses many long paragraphs and is difficult to follow. The authors should organize the introduction by research motivation, research design, incremental contributions, and break down long paragraphs into smaller ones.

Secondly, the paper lacks the necessary robustness test, which has verified that the conclusions of the paper are still valid after the change of research conditions. Otherwise, this conclusion is just a wonderful coincidence.

Reviewer 2 Report

This study is an empirical study on the influence of energy saving and emission reduction liability insurance pricing on the solvency of enterprises. I suggest the authors refine the academic contribution for this study. Also, there are a list of other remarks:

1. The article presents the correct approach to the research problem and contains new elements that constitute a good basis for publication in International Journal of Environmental Research and Public Health. Managerial implications formulated at the end are particularly important, but they should be more focused on the practical application of the method in real conditions.

2. Introduction: please add some final phrases in which is claimed the goals of the paper, the results expected, and a description of the content of the other sections.

3. Authors should discuss better the possible application of the method to the real case. Calculation complexity and time-consumption of the result obtaining should be better presented.

Based on this general assessment, I propose a minor revision for this manuscript. It can be expected that a fine paper is obtained in the end.

Round 2

Reviewer 1 Report

The paper has been revised according to the comments of the first round of review. On the one hand, the quality of the introduction has been significantly improved; On the other hand, the robustness of the research conclusions has also been improved.